# Mesenchymal Stem Cell Sheet Centrifuge-Assisted Layering Augments Pro-Regenerative Cytokine Production

**DOI:** 10.3390/cells11182840

**Published:** 2022-09-12

**Authors:** Sophia Bou-Ghannam, Kyungsook Kim, Makoto Kondo, David W. Grainger, Teruo Okano

**Affiliations:** 1Cell Sheet Tissue Engineering Center (CSTEC), Department of Pharmaceutics and Pharmaceutical Chemistry, Health Sciences, University of Utah, Salt Lake City, UT 84112, USA; 2Department of Biomedical Engineering, University of Utah, Salt Lake City, UT 84112, USA; 3Institute of Advanced Biomedical Engineering and Science, Tokyo Women’s Medical University, Tokyo 162-8666, Japan

**Keywords:** three-dimensional tissue, scaffold-free tissue, cell therapy, tissue engineering, regenerative medicine

## Abstract

A focal advantage of cell sheet technology has been as a scaffold-free three-dimensional (3D) cell delivery platform capable of sustained cell engraftment, survival, and reparative function. Recent evidence demonstrates that the intrinsic cell sheet 3D tissue-like microenvironment stimulates mesenchymal stem cell (MSC) paracrine factor production. In this capacity, cell sheets not only function as 3D cell delivery platforms, but also prime MSC therapeutic paracrine capacity. This study introduces a “cell sheet multilayering by centrifugation” strategy to non-invasively augment MSC paracrine factor production. Cell sheets fabricated by temperature-mediated harvest were first centrifuged as single layers using optimized conditions of rotational speed and time. Centrifugation enhanced cell physical and biochemical interactions related to intercellular communication and matrix interactions within the single cell sheet, upregulating MSC gene expression of connexin 43, integrin β1, and laminin α5. Single cell sheet centrifugation triggered MSC functional enhancement, secreting higher concentrations of pro-regenerative cytokines vascular endothelial growth factor (VEGF), hepatocyte growth factor (HGF), and interleukin-10 (IL-10). Subsequent cell sheet stacking, and centrifugation generated cohesive, bilayer MSC sheets within 2 h, which could not be accomplished within 24 h by conventional layering methods. Conventional layering led to H1F-1α upregulation and increased cell death, indicating a hypoxic thickness limitation to this approach. Comparing centrifuged single and bilayer cell sheets revealed that layering increased VEGF production 10-fold, attributed to intercellular interactions at the layered sheet interface. The “MSC sheet multilayering by centrifugation” strategy described herein generates a 3D MSC-delivery platform with boosted therapeutic factor production capacity.

## 1. Introduction

Clinical efforts have increasingly shifted toward exploiting mesenchymal stem cells (MSCs) for their paracrine-centric therapeutic mechanism, whereby MSCs secrete a broad range of bioactive cytokines and growth factors that can stimulate nearby cells via paracrine signaling [1,2,3]. This MSC paracrine effect guides tissue regeneration by exerting immunomodulation and stimulating progenitor cell angiogenesis, proliferation, and differentiation in the target tissue [4,5,6], and has been therapeutically utilized for applications in wound healing [7,8], myocardial repair [9,10], and liver fibrosis mitigation [11]. MSC therapeutic applications are inherently improved by engineering MSCs as 3D constructs, such as cell-seeded scaffolds or hydrogels and self-aggregated spheroids, that can be directly engrafted to injured tissue and resolve spatial and temporal cell retention, graft survival, and localized maintenance of secreted factors [12,13,14,15]. More recently, numerous cell–matrix and cell–cell interaction proteins were reported to be preserved in 3D MSC culture, in addition to promoting physiologic phenotypes, are responsible for measured augmentation of MSC paracrine function [16,17,18]. 

Cell sheet tissues are engineered using commercial, temperature-responsive polymer-grafted cell culture dishes (TRCD), on which cells are grown to confluence under adherent conditions, endogenously depositing ECM and forming adhesion interactions and junctions with the ECM and with adjacent cells. Aqueous temperature reduction from 37 °C to 20 °C prompts a surface property change from hydrophobic to hydrophilic that releases the confluent cell layer from the culture surface [19,20], inducing spontaneous contraction of the released cell layer that reorganizes the interconnected cell sheet into a 3D tissue comprised entirely of cells and endogenous matrix [21,22]. The focal advantage of cell sheet technology has been superiority as a 3D cell-delivery platform: being stably biologically adhesive without requiring suturing [23,24,25,26,27], absent any interruption of transplanted cells and host tissue communication commonly caused by biomaterial scaffolds or encapsulations [28,29,30], and supportive of sustained cell engraftment, survival, and reparative function [31]. Recently, it has been demonstrated that the intrinsic cell sheet 3D tissue-like microenvironment conditions the constituent MSCs, stimulating pro-regenerative cytokine gene expression and production in response to 3D organization that increases cell–cell and cell–matrix interactions relative to MSCs cultured as 2D adherent monolayers [21].

Building on previous assessments of cellular interactions in 2D and 3D cell sheets and their role in boosting MSC cytokine production potency compared to single cell formulations [21,32], and findings that bone marrow MSC sheet layering facilitated matured cartilage formation [33], this study hypothesized that MSC sheet paracrine properties could be further enhanced by sheet multilayering using centrifugation to non-invasively increase cellular interactions related to MSC paracrine function. Centrifugal force has previously been applied to myocardial sheets to facilitate immediate adherence between layers and stimulate cellular communications related to cardiac function, and, similarly, centrifugal force is commonly used for MSC scaffold-free chondrogenesis to imitate the condensation process and promote critical cellular and matrix interactions [34,35,36,37]. While cell sheet centrifugation has been extensively employed to construct multilayered functional tissues in vitro to replace diseased tissue in vivo [38,39,40,41,42], centrifuge manipulation of 3D cellular interactions to stimulate intrinsic MSC cytokine production in cell sheets has not been previously investigated. This study presents a comparison of centrifuged and conventional MSC sheets, both as single- and multi-layer 3D MSC sheets, to distinguish impact on 3D structure relationships to cell physical and biochemical interactions. MSC sheet paracrine cytokine signal production in response to centrifugation and multilayering is assessed over time in vitro.

## 2. Materials and Methods

### 2.1. Human Umbilical Cord Mesenchymal Stem Cell (hUC-MSC) Culture

Banked hUC-MSCs (Jadi Cell LLC, Miami, FL, USA) were thawed and seeded at a density of 4500 cells/cm^2^ and expanded in growth medium containing Dulbecco’s Modified Eagle’s Medium (DMEM) (Life Technologies, Carlsbad, CA, USA) supplemented with 10% fetal bovine serum (FBS) (Thermo Fisher Scientific, Waltham, MA, USA), 1.0% penicillin streptomycin (PS) (Gibco, Grand Island, NY, USA), 1.0% Glutamax (Life Technologies), and 1.0% non-essential amino acids (Life Technologies) and incubated in a humidified environment (37 °C, 5.0% CO_2_). Growth medium was exchanged after 24 h of initiating culture and every 2 days subsequently. hUC-MSCs were sub-cultured upon reaching 85% confluence.

### 2.2. hUC-MSC Sheet Fabrication

Passage 5 hUC-MSCs were sub-cultured using 0.05% Trypsin-EDTA (Gibco) and the cell suspension density was measured by a trypan blue exclusion assay. The resultant passage 6 hUC-MSCs were resuspended in growth medium supplemented with 20% FBS and 50 μg/mL L-ascorbic acid 2-phosphate (Sigma-Aldrich, St. Louis, MO, USA). P6 hUC-MSCs were seeded at 41,580 cells/cm^2^ onto 35 mm diameter UpCell™ temperature-responsive culture dishes (TRCDs) (CellSeed, Tokyo, Japan) and cultured for 4 days in a humidified environment without exchanging medium. At 4 days, confluent hUC-MSCs on TRCDs were moved to 20 °C and spontaneously detached from TRCD surfaces within 30 min, generating cell sheets.

### 2.3. Cell Sheet Tilting Optimization to Remove Excess Culture Media

Excess medium at the cell sheet-to-surface interface could inhibit stable cell sheet adherence upon centrifugation (Appendix A). Therefore, cell sheet tilting was optimized to remove excess culture medium at the cell sheet and culture surface interface without cell dehydration. Immediately following detachment, cell sheets were transferred onto pre-coated (100% FBS overnight), 1.0 μm-diameter pore size 6-well cell culture insert membranes (Falcon, Corning, NY, USA). Excised from the insert wells, cell sheets on insert membranes were transferred to the centers of 35-mm tissue culture plastic (TCP) dishes (CELLTREAT, Pepperell, MA, USA). Cell sheets on the insert membrane/TCP dish were tilted upright at an approximate 45 angle for 1, 5 or 10 min to remove excess culture medium. Afterward, PBS dyed with 0.4% trypan blue solution (Cell Culture Tested Trypan Blue Solution, Sigma Aldrich) was added to the center of the cell sheet and assessed for excess media presence (i.e., presence of blue solution beyond the cell sheet perimeter) or cell sheet dehydration (i.e., blue solution concentrated on the cell sheet surface) (*n* = 3 sheets per condition). 

### 2.4. Single Layer Cell Sheet Centrifugation

Cell sheets detached from TRCD by temperature reduction were transferred to 16-h FBS-coated insert membranes on TCPs and tilted for 5 min. TCPs containing cell sheets were then placed into a polydimethylsiloxane (PDMS) mold designed to secure the TCP in the center of a 6-well plate lid [38,39,40], and this lid was loaded, with balance, into a swing-type plate centrifuge with 16 cm rotor radius (Eppendorf) set to 37 °C internal temperature. The 1-layer cell sheets were centrifuged at 114× *g* (g-force, or relative centrifugal force) for 2 min (see Appendix A and Data for rationale). For comparison, 1-layer cell sheets were generated without centrifugation (i.e., conventional method [39]).Briefly, single cell sheets were transferred to 16-h FBS-coated insert membranes on TCPs and tilted for 5 min, then incubated for 1 h (37 °C, 5.0% CO_2_) to achieve adherence with the culture surface. For further analysis, centrifuged and conventional 1-layer cell sheets were cultured in 10% FBS medium for 1 h or 24 h after fabrication. 

### 2.5. Preparation of Layered Cell Sheets by Conventional and Centrifugation Methods

A flow chart of the cell sheet-layering protocols utilized in this study is summarized in Figure 1. Immediately following detachment from TRCDs by temperature reduction, the first-layer cell sheets were transferred to 16-h FBS-coated insert membranes on TCPs and tilted for 5 min. Using the conventional layering method, described previously by Haraguchi and colleagues [38,39,40], sheets were then incubated for 1 h (37 °C, 5.0% CO_2_) to adhere with the culture surface. A second detached cell sheet was moved onto the first layer cell sheet using warmed growth medium, spread over the first layer surface by medium aspiration, and tilted for 5 min. The resulting 2-layer cell sheet was incubated at 37 °C for 1 h to achieve adherence between the layered cell sheets.

The centrifugation layering method implemented herein is adapted from previously developed protocols by Haraguchi, Shimizu, and colleagues [38,39,40,42,43]. By the centrifugation method, the first-layer cell sheets were transferred to FBS-coated insert membranes on TCPs, tilted for 5 min, and centrifuged, per the above protocol described for single layer sheet centrifugation, (37 °C) at 114× *g* for 2 min. Afterward, the second-layer cell sheet was spread over the first-layer cell sheet by medium aspiration, tilted for 5 min, and incubated at 37 °C for 30 min before centrifuging the 2-layer sheets at 114× *g* for 2 min at 37 °C. The total time required for 2-layer cell sheet fabrication was 130 min or 44 min using the conventional or centrifugation method, respectively. For further analysis, 2-layer cell sheets were cultured in 10% FBS medium for 1 h or 24 h after layering. To control for the influence of cell sheet tissue-like structure and interactions on MSCs, single MSCs as cell suspensions were prepared for comparison. Passage 6 hUC-MSCs were seeded with the same density as cell sheets onto 60-mm diameter TCP dishes for 24 h. Prior to cell-to-cell contact, single MSCs were harvested as cell suspensions from the dish using TRIzol (Ambion, Life Technologies, Carlsbad, CA, USA). 

### 2.6. Histological Analysis

To analyze the structure of layered cell sheets, 1-layer and 2-layer cell sheets fabricated by conventional, or centrifugation methods were fixed with 4.0% paraformaldehyde (PFA) (Thermo Fisher Scientific) for 30 min and paraffin embedded. Embedded samples were sectioned at 4.0 μm and stained with Mayer’s Hematoxylin (Sigma-Aldrich) and Eosin (Thermo Fisher Scientific) (H&E) to visualize the cell sheet dimensions in cross-section. Stained cell sheet sections were dried overnight and imaged with a BX41 widefield microscope (Olympus, Tokyo, Japan) using AmScope Software (AmScope v4.8.15934, Irvine, CA, USA). To calculate 1- or 2-layer cell sheet thicknesses, 5 H&E-stained images were captured along the length of 24 h cultured samples (*n* = 3 per group), and 5 linear measurements from the apical to basal plane were made per image using AmScope Software (AmScope v4.8.15934); these 75 linear measurements of thickness were averaged per group.

### 2.7. Cell Proliferation Rate

The absolute cell number in 1- and 2-layer cell sheet samples was quantified 24 h after layering using a tissue destructive method modified from previous reports [44]. Briefly, culture medium was removed, and each sample was rinsed twice with PBS. Trypsin-EDTA (0.25%, 2 mL) (Gibco) was added directly to the samples and incubated at 37 °C first for 10 min in a humidified incubator, then for 15 min in a 37 °C water bath. Afterwards, trypsin was removed from the cells by centrifugation (258× *g*, 5 min) and supernatant aspiration. Cell pellets were dispersed with 0.5 mL collagenase P (0.25 mg/mL, Roche, IN, USA) and incubated for 10 min in a 37 °C water bath. At this point, cell sheets were fully digested into single cell suspensions. An additional 0.5 mL of cell growth medium was added to each cell suspension to total 1.0 mL, and exact cell numbers and population viability were measured using a trypan blue exclusion assay (*n* = 3 sheets per group). Table 2 reports live cell numbers as 1×10^4^ cells. Viable cell numbers were not measurable in the 2-layer conventional cell sheet group due to early cell death that resulted in cell sheet detachment prior to day 4 in culture. The proliferation rate over 24 h was measured as the fold increase in live cell number from 1 h to 24 h in 1-layer conventional and centrifuged cell sheet layered groups. 

### 2.8. Reverse Transcription Quantitative Polymerase Chain Reaction (RT-qPCR)

Total RNA was isolated from cell sheet samples (*n* = 3 sheets per group) 1, 2 or 4 days after layering, and from single MSC cell suspension samples (*n* = 4) at 1 day after seeding, using TRIzol (Ambion, Life Technologies) with the PureLink 18 RNA Mini Kit (Invitrogen, Thermo Fisher Scientific, Waltham, MA, USA) according to manufacturer instructions. Isolated RNA was quantified with a NanoDrop Spectrophotometer (Thermo Fisher Scientific) and all cDNA samples were prepared from 1.0 μg of RNA/sample using a high-capacity cDNA reverse transcription kit (Life Technologies). Genes were quantified using quantitative PCR with commercially available Applied Biosystems primers (glyceraldehyde 3-phosphate dehydrogenase [GAPDH, Hs99999905_m1] as a housekeeping gene, β-catenin [Hs00355049_m1], integrin β1 [Hs01127536_m1], connexin 43 [Hs04259536_g1], laminin α5 [Hs00966585_m1], VEGFA [Hs99999070_m1], HGF [Hs00379140_m1], IL-10 [Hs00961622_m1], β-actin [Hs99999903_m1], and hypoxia inducible factor 1 alpha [HIF-1α, Hs00153153_m1]), and was performed on Step One Plus (Applied Biosystems, Foster City, CA, USA). Relative gene expression was determined using the comparative threshold cycle (*C*_T_) change algorithm. 

### 2.9. Soluble Cytokine Secretion Quantification

First, 1-layer and 2-layer cell sheets fabricated by conventional or centrifugation methods were transferred to 6-well cell culture inserts and cultured with 5 mL of fresh growth medium for 4 days (*n* = 3 sheets per group). During the culture, the media were changed every 24 h and the supernatants were collected at each medium change. At the end of culture, the cell sheets were digested, and live cell numbers were counted in the same manner as previously described (vide supra). Concentrations of soluble VEGF, HGF, and IL-10 were quantified in the collected supernatants using human VEGF, human HGF, and human IL-10 Quantikine ELISA kits (R&D Systems, Minneapolis, MN, USA), respectively, according to manufacturer’s instructions. To determine the concentration of cytokines secreted per cell, concentration values at each time point were normalized by the average live cell number counted at Day 1 (Table 2).

### 2.10. Statistical Analysis

All statistical analysis was conducted on data sets of *n* ≥ 3 biological replicates, with quantitative values expressed as a mean ± standard error (SE). All data sets were evaluated for normality using a Shapiro–Wilk test (GraphPad Prism 9 software: Prism 9.0.0, https://www.graphpad.com/scientific-software/prism/, accessed on 30 July 2022, San Diego, CA, USA). Comparisons of two groups were tested using a two-tailed, unpaired Student’s *t*-test. Statistical significance between three or more groups was tested using a one-way analysis of variance (ANOVA) with a Tukey test correction for multiple comparisons. Statistical significance was defined as * *p* < 0.05, ** *p* < 0.01, and *** *p* < 0.001. No statistical significance was defined as *p* ≥ 0.05. Statistical analysis was conducted using GraphPad Prism 9 software (Prism 9.0.0).

## 3. Results

### 3.1. Optimization of Cell Sheet-to-Surface Interactions

Excess medium at the cell sheet–insert membrane interface could inhibit stable cell sheet adhesion to the culture surface following centrifugation. The duration of passive, cell sheet tilting (Figure 2a) was therefore optimized to remove excess interfacial medium without dehydrating the cells. In the case of 1 min tilting, the trypan blue-dyed solution seeped past the cell sheet perimeter, indicating medium still existed at the cell sheet–insert membrane interface (Figure 2b). For 10-min tilting, the trypan blue-dyed solution concentrated on the cell sheet surface, indicating that the cell sheet had drained this excess medium from the culture-ware and was visually dehydrated (Figure 2d). After 5-min tilting, the trypan blue-dyed solution spread to the cell sheet perimeter, indicating that excess interfacial medium had been cleared without drying the cell sheet (Figure 2c). From these findings, 5-min tilting was implemented for the centrifugation protocol. 

### 3.2. Centrifugation Alters Cellular Structures

Cell sheet adherence to the culture surface under a range of centrifugation speeds and durations was evaluated quantitatively by a mechanical rotation test. Centrifugation at 29× *g* and 114× *g* required 120 s to maintain sheet adherence in all three trials, failing to achieve 100% adherence at 60, 30, 15, or 5 s (Table 1 and Appendix A). Increasing centrifugation speed to 458× *g* and 1030× *g* consistently maintained cell sheet surface adherence for 60 and 30 s, with successful adherence for 15 and 5 s in all three trials at 1030× *g* (Table 1 and Appendix A). Centrifugation at 1832× *g* displayed visible deformation due to sheet sliding within 5 s, considered a failure for stable cell sheet centrifugation (Table 1 and Appendix A). 

Histological images of conventional and centrifuged (114× *g*, 2 min) one-layer cell sheet cross-sections stained with H&E are shown in Figure 3. Within 1 h of fabrication, conventional cell sheets showed a loose cell packing arrangement and an inconsistent apical tissue surface (Figure 3a). Conversely, centrifuged cell sheets showed a visibly tight-packed physical arrangement; the apical tissue surface was level and flat, rendering a consistent thickness across the tissue (Figure 3c). The cross-sectional tissue structure of centrifuged cell sheets (Figure 3d) was 1.8-fold decreased in thickness (Figure 3e) compared to conventional cell sheets (Figure 3b) (22 ± 3.2 μm and 41 ± 7.3 μm, respectively (*p* < 0.0001)) 24 h after fabrication. No significant difference in cell proliferation over 24 h was measured between one-layer conventional and centrifuged cell sheet groups (Figure 3f). Potential hypoxic tissue conditions were evaluated by relative gene expression of HIF-1α per MSC at 24 h. No significant upregulation of HIF-1α in centrifuged cell sheets was measured (Figure 3g). Additionally, the population percentage of dead cells was not significantly different between conventional and centrifuged cell sheets at 24 h (Figure 3h).

### 3.3. Centrifugation Enhances Gene Expression Related to Cellular Interactions in One-Layer Cell Sheets

To demonstrate the cell sheet “tissue effect” [21] and impact of centrifugation, relative gene expression of cell–cell and cell–matrix interaction proteins was compared among single cell suspension, conventional, and centrifuged one-layer sheets 24 h after fabrication. Gene expression for β-catenin, that intracellularly binds extracellular cadherin to mediate cell adhesion between adjacent cells [45,46], was significantly decreased by centrifugation relative to conventional cell sheets (*p* = 0.023), though conventional sheet fabrication alone significantly upregulated β-catenin relative to the single cell control (Figure 4a). Gene expression for connexin 43, a cross-membranal gap junction protein that forms between physically adjacent cells and facilitates direct intracellular molecular signaling exchange [47], was significantly upregulated in centrifuged cell sheets relative to conventional cell sheets (*p* = 0.020) and single cells (Figure 4b). Gene expression for integrin β1, a cell surface receptor that extracellularly binds ECM ligands [48], and, concomitantly, laminin α5, a cell-adhering ECM glycoprotein [49], was significantly upregulated by cell sheet centrifugation (*p* = 0.0011 and *p* = 0.011, respectively) relative to the single cell control (Figure 4c and Figure 4d, respectively). These data suggest that one-layer cell sheet centrifugation promotes biochemical interactions related to cell–cell and cell–matrix adhesion per MSC, corresponding with increased physical interactions due to the observed tight-packed tissue structure. 

### 3.4. Enhanced Tissue Interactions Due to Centrifugation Increases MSC Pro-Regenerative Cytokine Production

The impact of centrifuge-induced physical and biochemical interactions on MSC function was evaluated by comparing one-layer cell sheet cytokine production. Vascular endothelial growth factor (VEGF), hepatocyte growth factor (HGF), and interleukin-10 (IL-10) gene expression per MSC are significantly upregulated in one-layer centrifuged cell sheets relative to conventional cell sheets 24 h after fabrication (*p* = 0.018, *p* = 0.036, and *p* = 0.0047, respectively) and the single cell control (Figure 4e–g, respectively). In the analysis of culture supernatants, MSCs in one-layer centrifuged cell sheets secreted significantly more VEGF, HGF, and IL-10 per viable cell at each time point in culture (IL-10 increased on average at 3 days) (Figure 4h–j, respectively) compared to MSCs in one-layer conventional cell sheets. Average viable cell number at 4 days was not significantly changed from that measured at 1 day in conventional or centrifuged one-layer cell sheets (*p* = 0.77 and *p* = 0.22, respectively), tested using a two-tailed, unpaired Student’s *t*-test (Table 2). These data show that individual MSC cytokine production function is augmented by cell sheet centrifugation. 

### 3.5. Comparative Assessment of Two-Layer Cell Sheet Structure and Viability Fabricated by Conventional and Centrifugation Layering Methods

Histological assessment of two-layer cell sheets fabricated by the conventional method (H&E stained, visualized as cross-sections) revealed that, although adhered in medium, layered sheets did not interface within 1 h following layering (Figure 5a), but the sheet–sheet interfacing was achieved in 24 h of culture (Figure 5b). Conversely, centrifugation immediately interfaced layered cell sheets (Figure 5c) to generate a compacted, homogenous tissue structure by 24 h (Figure 5d). Analysis of cross-sectional tissue structure in 1-h samples found conventionally layered sheets significantly thicker than those layered by centrifugation (84 ± 6.8 μm and 50 ± 1.8 μm, respectively (*p* < 0.0001)) (Figure 5e). The population percentage of dead cells in two-layer conventional cell sheets was higher than that in two-layer centrifuged cell sheets (Figure 5f). HIF-1α was significantly upregulated at 24 h relative to two-layer centrifuged sheets (*p* = 0.0001) (Figure 5g). 

### 3.6. Centrifugation Enhances Cellular Function of Layered Cell Sheets

To investigate two-layer sheet cellular function, gene expression levels related to cell–cell interaction, cell–ECM interaction, and regenerative cytokine production were measured. MSC gene expression for β-catenin was significantly upregulated in two-layer conventional cell sheets (*p* = 0.018) (Figure 6a). Sheets fabricated by centrifugation layering demonstrated significantly increased MSC gene expression for a gap junction protein, connexin 43 (*p* = 0.0067), as well as for cell–ECM interaction-related proteins, integrin β1 (*p* = 0.0048) and laminin α5 (*p* = 0.0003) (Figure 6b–d, respectively), relative to MSCs in two-layer conventional cell sheets. VEGF, HGF, and IL-10 gene expression per MSC are significantly upregulated in two-layer centrifuged cell sheets 24 h after fabrication (*p* = 0.037, *p* = 0.015, and *p* = 0.021, respectively) (Figure 6e–g, respectively). Additionally, secreted VEGF, HGF, and IL-10 concentrations in culture supernatants were 1.3-fold, 1.8-fold, and 2.0-fold increased per cell (*p* = 0.011, *p* = 0.0001, and *p* = 0.0003, respectively) from two-layer centrifuged sheets relative to conventional sheets (Figure 6h–j). Live cell numbers were similar in two-layer conventional (126 × 10^4^ ± 7.0 × 10^4^ cells) and centrifuged sheets (123 ± 5.1 × 10^4^ cells) at 1 day in static culture. 

### 3.7. Cell Sheet Layering Augments MSC Cytokine Production 

Cell sheet layering introduces an interface of cross-sheet interactions. Although relative gene expression for β-catenin (Figure 7a), integrin β1 (Figure 7b), and laminin α5 (Figure 7c) were mostly similar over time, apart from β-catenin at day 4 (*p* = 0.0069), MSC gene expression for gap junction protein, connexin 43, was significantly upregulated in two-layer centrifuged cell sheets at 2 (*p* = 0.042) and 4 days (*p* = 0.042) after layering relative to one-layer centrifuged cell sheets (Figure 7d). To measure the impact of cell sheet layering on MSC cytokine production, culture supernatants from one-layer and two-layer centrifuged cell sheets were analyzed over 4 days. Normalized for viable cell number at day 1, MSCs in two-layer centrifuged cell sheets secreted 2.0-fold, 5.3-fold, 6.3-fold, and 2.9-fold more VEGF than one-layer centrifuged cell sheets at 1, 2, 3, and 4 days, respectively (1 day: *p* = 0.0003, 2 days: *p* = 0.0002, 3 days: *p* = 0.0132, and 4 days: *p* = 0.0041) (Figure 7e). HGF and IL-10 secretions per MSC in two-layer centrifuged sheets remained similar to MSCs in the one-layer condition over 4 days (Figure 7f,g). 

## 4. Discussion

The aim of this study was to stimulate intrinsic MSC cytokine production potency through cell sheet engineering to enhance clinically useful MSC paracrine effects for applications in regenerative medicine. A previous study demonstrated the importance of physical, and cellular communications interactions in a 2D cell sheet monolayer on upregulating MSC paracrine capacity relative to a dissociated MSC formulation [32], and, further, an additional study found that upon spontaneous cell sheet contraction into a 3D tissue with higher abundance cellular interactions, MSC paracrine capacity was similarly increased, measured as in vitro pro-regenerative cytokine production per cell [21]. The present study explores for the first time the feasibility of centrifugation as a tool to non-invasively increase cell–cell physical encounters and biochemical cellular interactions related to cytokine secretion within MSC sheets. 

Serum containing cell attachment factors was used to pre-coat insert membranes and support cell sheet adhesion by binding intact matrix receptors [50,51], demonstrating that 16-h precoating prevented tissue deformation under centrifugal force. Additionally, cell sheet-to-surface adhesion requires direct interfacing, yet sheets were cultured, harvested, and manipulated onto the culture surface in aqueous conditions, interstitially trapping residual medium that buffers direct contact. Higher g-forces could expel interfacial medium, but risk cell damage and sheet deformation. Therefore, prior to centrifugation, cell sheets on dishes were tilted upright (approximately 45°) to passively expel medium from the sheet–surface interface (Figure 2a) [52]. Tilting for 5 min was sufficient for media expulsion while maintaining sheet hydration (Figure 2c). Additionally, sheet centrifugation at 114× *g* for 120 s (2 min) was selected for consistent and sufficient cell sheet adhesion to the culture surface in media, withstanding displacement under mechanical rotation testing (Table 1 and Appendix A). 

Centrifugation of single layer cell sheets has previously been implemented to compact cell sheets into a more ordered structure with greater cell–cell physical contact that, in turn, could better propagate molecular signaling throughout cell sheets [38,39,40]. Enhanced MSC regenerative potential using cell sheet centrifugation was hypothesized to result from higher abundance cell-to-cell physical and biochemical interactions, boosting MSC cytokine production important to tissue repair and regeneration. Indeed, centrifugation increased physical cell contact: histological analysis of one-layer centrifuged cell sheets revealed a denser cell packing arrangement relative to one-layer conventional cell sheets 1 h after fabrication (Figure 3a,c), consistent with a measured 1.8-fold relative reduction in sheet thickness due to centrifugation by 24 h (Figure 3b,d,e). No differences in gene expression of HIF-1α, a transcriptional regulator of the adaptive response to hypoxia [53], or in cell death at 24 h confirmed that this near halving in centrifuged tissue thickness was not due to hypoxic limitations or cell loss due to cell death (Figure 3g,h). Cell shape change, regulated by the actin cytoskeleton [54], did not appear to be a major contributing factor in centrifuged tissue compaction (Figure 3i). β-catenin regulates cadherin-mediated cell adhesion and the migratory ability of cells [45,46]; the measured decrease in centrifuged sheet β-catenin gene expression further evidence inhibited cell migratory potential in the tight-packed tissue structure with high intercellular adhesive tension (Figure 4a). Importantly, gene expression for proteins that mediate the biochemical interaction of neighboring cells, namely connexin 43, a cross-membranal gap junction protein [47], integrin β1, an extracellular ECM binding protein [48], and laminin α5, an integrin-receptor ligand [49], were significantly upregulated due to one-layer sheet centrifugation (Figure 4b–d). Conceivably, increased physical cell contacts within the centrifuged sheet triggered subsequent molecular interactions, as has similarly been observed with centrifuged cardiac cell sheets [38].

Enhanced cellular interactions within 3D tissues regulate MSC cytokine production potency. For instance, MSC upregulation of gap junction protein formation has previously been attributed to boosting MSC VEGF production, promoting angiogenesis [55]. MSC HGF production is significantly increased with facilitated cell–cell interactions [16]. Moreover, tissue-like 3D culture that enabled cell–ECM interactions upregulated MSC IL-10 production relative to non-matrix-interacting MSCs [56]. Therefore, MSC production of VEGF, HGF, and IL-10, cytokines with specific implications in vascularization [57], fibrosis mitigation [11], and inflammation mediation [58], respectively, by conventional and centrifuged cell sheets was assessed. Centrifugation significantly increased gene expression of VEGF, HGF, and IL-10 per cell relative to conventional sheets in 24-h samples (Figure 4e–g). Concomitantly, centrifuged sheets secreted significantly higher concentrations of VEGF, HGF, and IL-10 per viable MSC (Table 2) over 4 days in static culture (Figure 4h–j). Centrifugation, therefore, provides an external tool to promote and upregulate physical, and cell–cell communication interactions related to MSC cytokine production capacity within a cell sheet.

The comparative analysis of single cells against single-layered conventional and centrifuged cell sheets in Figure 4 highlights an apparent tissue effect on MSC function, whereby externally applied physical force induced by centrifugation enhanced 3D interfacial contact between cell–cell and cell–matrices, facilitating biochemical interactions that promote individual MSC cytokine production. Enzymatically dissociated MSCs are stripped of physical and protein-mediated contacts between neighboring cells and matrix (i.e., single cells) [32], while cell sheet engineering preserves confluent cellular interactions and endogenous MSC-derived ECM within a 3D tissue-like structure [21,32,59]. Figure 4a–g demonstrates enhanced MSC cytokine production capacity due not only to cell sheet tissue-like structure, but also to augmented tissue interactions by sheet centrifugation, relative to single MSCs as cell suspensions.

Operating from the hypothesis that the 3D tissue structure of centrifuged sheets, characterized by increased cell–cell and cell–matrix interactions, improves MSC cytokine-production potency, cell sheet layering was subsequently explored as a well-documented mechanism for controlling cellular interaction dynamics within cell sheet tissue [38,39,40,41,42,52]. Building from conditions for first-layer cell sheet adhesion, layering protocols using centrifugation (centrifugation method) or using passive incubation (conventional method [39]) were developed to support sheet-to-sheet attachment (Figure 1). Data in Figure 5 show that the centrifugation method generated a cohesively interfaced, layered cell sheet within 1 h that is not achieved by the conventional method, though both methods produced interfaced tissue by 24 h (Figure 5a–d). Although it has been well documented that hypoxic conditions are not directly related to cell survival [60,61], in this study, MSC viability declined in two-layer cell sheets fabricated by the conventional method, corresponding with significant upregulation of HIF-1α gene expression by 24 h (Figure 5e–g). These findings suggest that an oxygen diffusion threshold had been exceeded by conventional layering, which measured around 84 µm within 1 h of fabrication, thereby limiting the construct thickness to approximately 55 µm after 24 h. Thickness limitations have previously been reported in studies using multilayered cell sheets: human endometrial-derived mesenchymal cell sheets in static culture experienced a thickness limitation around 40 µm [62]. Five-layer human skeletal muscle myoblast sheets (approximately 70–80 µm thick) showed signs of hypoxia that three-layer HSMM sheets (approximately 40–50 µm thick) did not [52]. In the present study, centrifugation circumvented this limitation, generating a two-layered tissue with overall thickness averaging 50 µm at 1 h and 44 µm at 24 h after centrifugation layering.

Consistent with the centrifugation effect observed in one-layer cell sheets, MSCs in two-layer centrifuged sheets demonstrated significantly higher gene expression for gap junction protein, cell–matrix interaction-related proteins, and pro-regenerative cytokines relative to MSCs in two-layer conventional cell sheets (Figure 6b–g). Moreover, two-layer centrifuged cell sheet MSCs secreted significantly higher concentrations of therapeutic factors, VEGF, HGF, and IL-10, within 24 h of fabrication than MSCs in two-layer conventional cell sheets (Figure 6h–j). Certainly, gene expression and cytokine production could be impaired due to hypoxic conditions impacting cell viability in two-layer conventional cell sheets. However, given the incremental increase in thickness and cell density (Table 2) due to sheet multilayering with centrifugation, we expect that the centrifugation effect observed in one-layer cell sheets is conserved in two-layer cell sheets, contributing to the higher reported MSC secretory function relative to two-layer conventional sheets. 

Figure 7 data demonstrate an apparent tissue effect related to cell sheet layering. Though HGF and IL-10 production was proportional to the cell number per sheet layer, VEGF secretions by two-layer centrifuged sheets were on the order of 3.2-fold, 8.7-fold, 10-fold, and 4.8-fold higher than the one-layer centrifuged sheets at 1, 2, 3 and 4 days, respectively (Figure 7e–g). This demonstrates that cell sheet layering can be modulated to stimulate MSC cytokine production potency. Indeed, this is consistent with previous findings that cell sheet layering improved the functional capacity of the constituent cells, for applications of gap junction formation and electrical signal transmittance in cardiac cells sheets [38,40,41] as well as chondrogenic differentiation and cartilage maturation in MSC sheets [33]. 

The results of this study highlight several key features of cell sheet centrifugation and layering that collectively generate highly cytokine-secretory MSC tissue: (1) MSC sheets are compacted by centrifugal force, contributing a tight cell packing arrangement that increases physical cellular interactions, leading to (2) higher abundance cell–cell and cell–matrix interactions that enhance MSC pro-regenerative potency. Additionally, (3) tissue compaction due to centrifugation enables MSC sheet layering without exceeding oxygen diffusion thresholds in static culture, supporting (4) inter-sheet cellular interactions that significantly augment MSC angiogenic factor secretion. These findings illustrate an evident tissue effect, whereby 3D tissue culture that promotes cell–cell interactions enhances MSC paracrine potency, and that can be efficiently modulated using the centrifugation-based cell sheet layering platform. 

## 5. Conclusions

This study demonstrated that cell sheet centrifugation increases cell-experienced 3D interactions in culture, augmenting individual MSC cytokine production relative to non-centrifuged sheets and suspended MSCs. Enhanced MSC phenotypic properties relevant to their utility in possible cell therapy and regenerative medicine (e.g., increased cytokine production in culture) result from both cell–cell engagement within individual MSC sheets, and cell sheet–cell sheet interfacial forces in MSC layered constructs, both enforced by centrifugation. A tissue effect operative in this 3D layered, dense layered cell sheet system emulates cell phenotypic changes seen in other 3D cultures, but without any introduced 3D scaffolding or supporting biomaterial matrix beyond the endogenous ECM produced by MSC sheets. Layered MSC sheets with increased physical and biochemical engagement therefore represent a new living cell therapy system with considerable performance advantages over conventional MSC cell cultures and MSCs delivered in biomaterials devices.

Based on these findings, future work will consider multi-layering cell sheets beyond two-layers as a means of introducing additional interfaces of cellular interactions to further enhance MSC paracrine function. The addition of layers must consider oxygen diffusion thresholds that limit tissue thickness in static in vitro culture. To navigate this limitation, further studies could implement cell sheets with smaller initial seeding densities (thinner tissues) to maximize the number of layers achieved before a hypoxic threshold is reached, or co-culture with microvasculature-producing endothelial cells [63,64,65] to provide an avenue for neovascularization for oxygen distribution throughout thick tissues. Overall, the cell sheet centrifugation layering platform can be readily implemented to generate scaffold-free, 3D MSC tissue-like constructs with enhanced paracrine-relevant secretory functions for applications in regenerative medicine.

## Figures and Tables

**Figure 1 cells-11-02840-f001:**
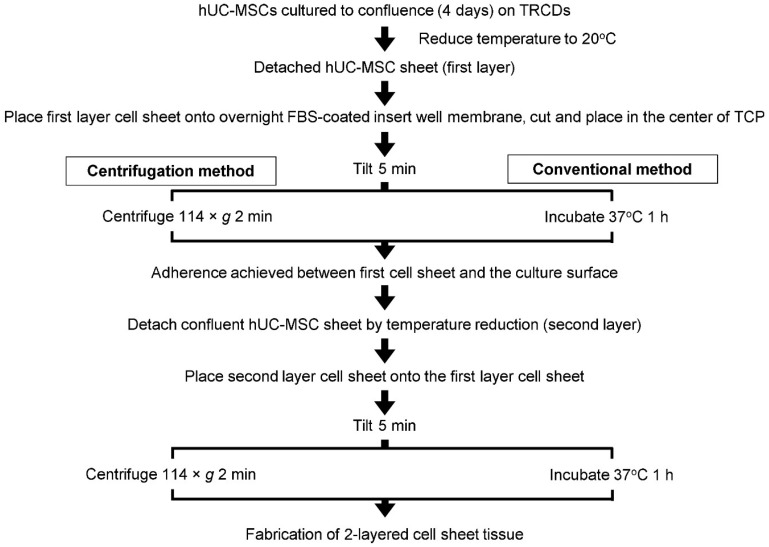
Flow chart illustrating the MSC sheet layering protocol with centrifugation (centrifugation method) and without centrifugation (conventional method).

**Figure 2 cells-11-02840-f002:**
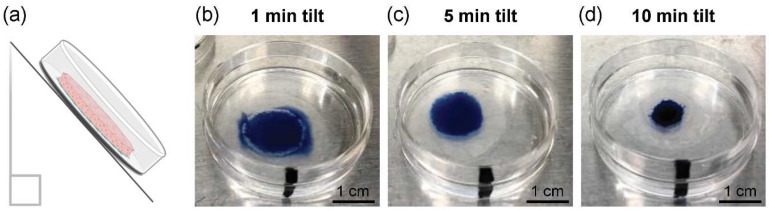
Single layer cell sheet surface interaction optimization. To remove excess interfacial culture medium without drying cell sheets, (**a**) cell sheets on insert membranes/TCP dishes were tilted upright at an approximate 45 angle for (**b**) 1 min, (**c**) 5 min or (**d**) 10 min. Scale bars = 1.0 cm.

**Figure 3 cells-11-02840-f003:**
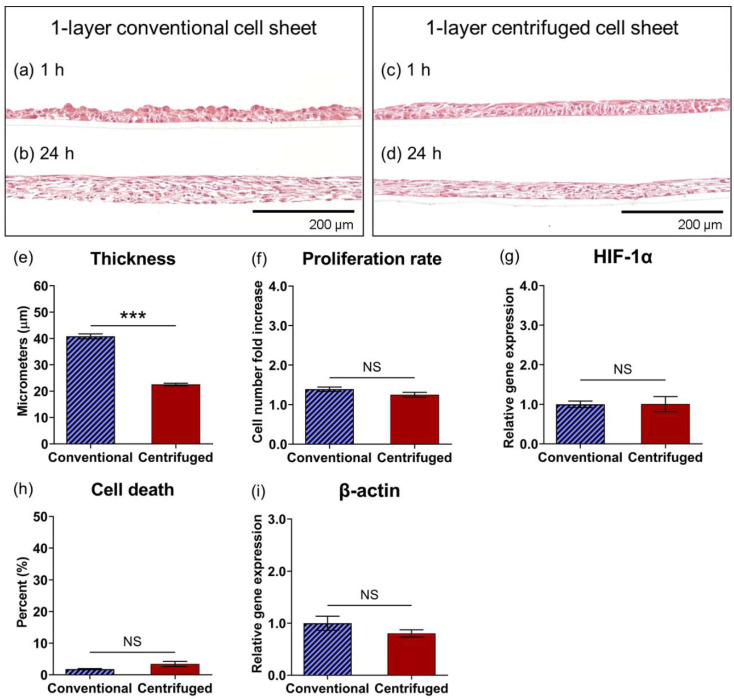
Centrifugation compacts cell sheet tissue structure. Histological cross-sections of conventional and centrifuged 1-layer cell sheets stained with H&E at (**a**,**c**) 1 h and (**b**,**d**) 24 h after fabrication, with quantified comparisons for 1-layer conventional and 1-layer centrifuged cell sheets (**e**) tissue thickness, (**f**) cell proliferation rate, (**g**) HIF-1α gene expression, (**h**) non-viable cell ratio, and (**i**) β-actin gene expression in 24-h samples. Gene expression is normalized to GAPDH and compared to the 1-layer conventional cell sheet. Scale bars = 200 μm. Values are means ± SE (*** *p* < 0.001). NS = not significant (*p* > 0.05).

**Figure 4 cells-11-02840-f004:**
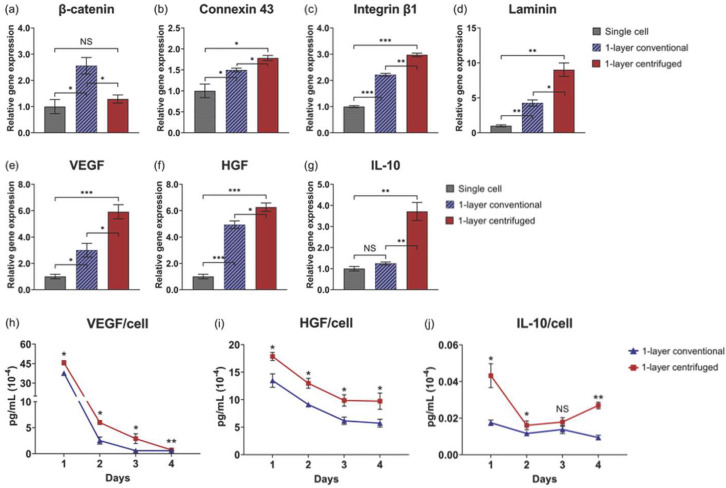
Cell sheet centrifugation enhances MSC pro-regenerative cytokine production related to cellular interactions. Quantitative gene expression of proteins related to cellular interactions, including (**a**) β-catenin (cell–cell interaction), (**b**) connexin 43 (gap junction), (**c**) integrin β1 (cell–ECM interaction), and (**d**) laminin α5 (ECM), and to cytokine production, including (**e**) VEGF, (**f**) HGF, and (**g**) IL-10, in conventional and centrifuged cell sheets at 24 h relative to the single cell suspension control. Analysis of cell sheet supernatants in static culture quantified (**h**) VEGF, (**i**) HGF, and (**j**) IL-10 secretion per sheet over 4 days, normalized to day 1 average viable cell number. Gene expression is normalized to GAPDH and compared to the single cell formulation of MSCs. Values are means ± SE (* *p* < 0.05, ** *p* < 0.01, and *** *p* < 0.001). NS = not significant (*p* > 0.05).

**Figure 5 cells-11-02840-f005:**
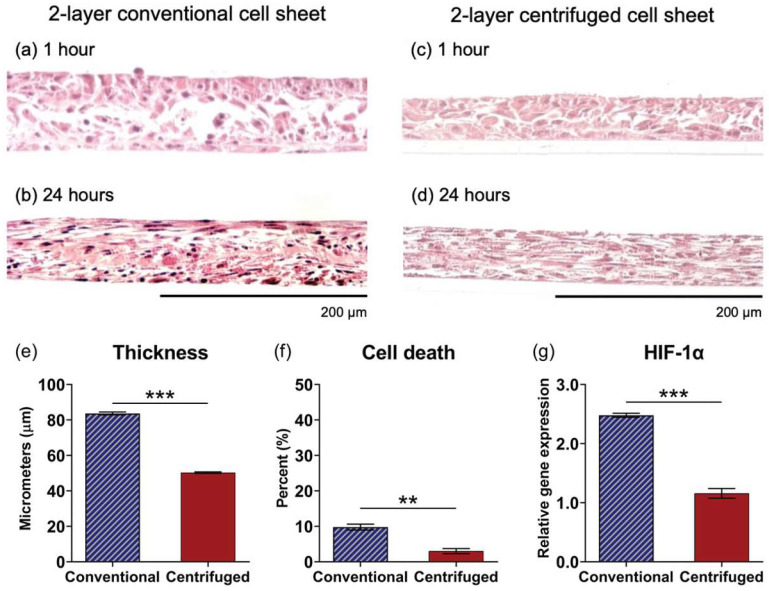
Centrifugation layering enables viable 2-layer cell sheet tissue fabrication. Histological assessment of 2-layer conventional and centrifuged cell sheets 1 h (**a**,**c**, respectively) and 24 h (**b**,**d**, respectively) after layering, visualized as cross-sections stained with H&E. Additionally, 2-layer cell sheets fabricated by conventional or centrifuge methods were compared for (**e**) tissue thickness in 1-h samples, and for (**f**) cell viability and (**g**) HIF-1α gene expression in 24 h samples. Gene expression is normalized to GAPDH and compared to the 1-layer conventional cell sheet. Scale bars = 200 μm. Values are means ± SE (** *p* < 0.01 and *** *p* < 0.001). NS = not significant (*p* > 0.05).

**Figure 6 cells-11-02840-f006:**
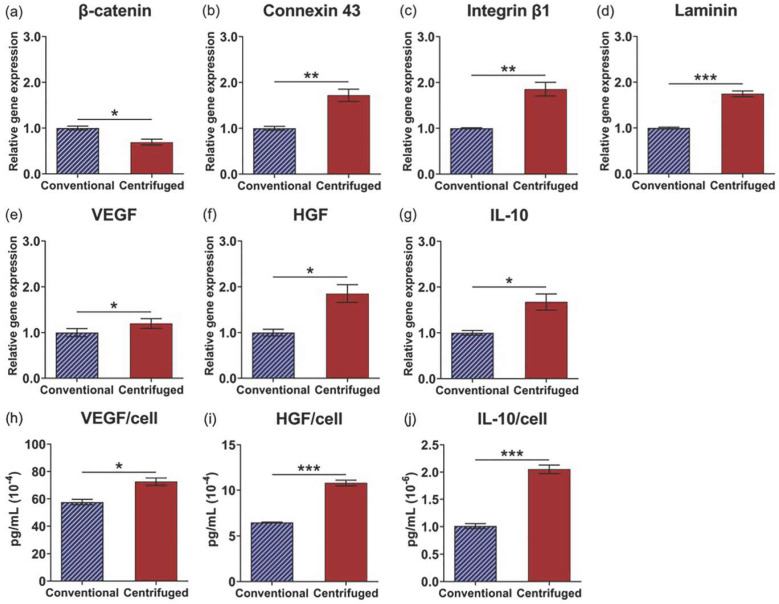
Centrifugation layering enhances MSC pro-regenerative cytokine production related to cellular interactions. Quantitative gene expression of cell interaction proteins (**a**) β-catenin, (**b**) connexin 43, (**c**) integrin β1, and (**d**) laminin α5 (ECM), and cytokines (**e**) VEGF, (**f**) HGF, and (**g**) IL-10 in 2-layer conventional and centrifuged cell sheets at 24 h. Analysis of cell sheet supernatants in static culture quantified (**h**) VEGF, (**i**) HGF, and (**j**) IL-10 secretion per 2-layer sheet over 24 h, normalized for average viable cell number. Gene expression is normalized to GAPDH and compared to the 2-layer conventional cell sheet. Values are means ± SE (* *p* < 0.05, ** *p* < 0.01, and *** *p* < 0.001).

**Figure 7 cells-11-02840-f007:**
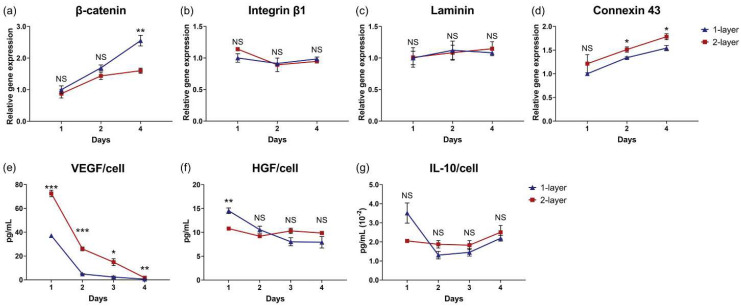
Cellular interactions introduced by cell sheet centrifugation layering augment MSC pro-regenerative cytokine production. Relative gene expression levels of (**a**) β-catenin, (**b**) integrin β1, (**c**) laminin α5, and (**d**) connexin 43 in 1-layer and 2-layer centrifuged cell sheets at 1, 2, and 4 days in culture. Analysis of 1-layer and 2-layer centrifuged cell sheet supernatants in static culture quantified concentrations of (**e**) VEGF, (**f**) HGF, and (**g**) IL-10 secreted per viable MSC over 4 days. Gene expression is normalized to GAPDH and compared to the 1-layer centrifuged cell sheet at 1 day. Values are means ± SE (* *p* < 0.05, ** *p* < 0.01, and *** *p* < 0.001). NS = not significant (*p* > 0.05).

**Table 1 cells-11-02840-t001:** Single layer cell sheet attachment rate following medium addition and mechanical rotation test.

	Time	5 s	15 s	30 s	60 s	120 s
×*g*						
29		0% (0/3 trials)	0% (0/3 trials)	0% (0/3 trials)	66% (2/3 trials)	100% (3/3 trials)
114		0% (0/3 trials)	0% (0/3 trials)	33% (1/3 trials)	66% (2/3 trials)	100% (3/3 trials)
458		33% (1/3 trials)	66% (2/3 trials)	100% (3/3 trials)	100% (3/3 trials)	--
1030		100% (3/3 trials)	100% (3/3 trials)	100% (3/3 trials)	--	--
1832		Sheet deformation	--	--	--	--

**Table 2 cells-11-02840-t002:** Cell sheet viable cell numbers in static culture.

	1 Day	4 Days
1-layer conventional cell sheet (1 × 10^4^ cells)	75 ± 9.5	77 ± 8.5
1-layer centrifuged cell sheet (1 × 10^4^ cells)	61 ± 6.6	72 ± 11
2-layer conventional cell sheet (1 × 10^4^ cells)	126 ± 7.0	*Not measurable*
2-layer centrifuged cell sheet (1 × 10^4^ cells)	123 ± 5.1	128 ± 15

## Data Availability

The data presented in this study are available from the corresponding author upon reasonable request.

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
