# Peer review of "Mesenchymal Stem Cell Sheet Centrifuge-Assisted Layering Augments Pro-Regenerative Cytokine Production"

_cells, 2022, doi:10.3390/cells11182840_

Round 1

Reviewer 1 Report

The aim of this study was the comparison of centrifuged and conventional MSC sheets to distinguish the impact on 3D structure relationships to cell physical and biochemical interactions, Some of the issues have to be reviewed. You can find the related comments below.

1.      Introduction

a.      Please add references applying the power of centrifuge to improve adherence of cells

b.      Why is necessary to increase the number of the cell sheet layer

2.      Material and methods

a.      What did you mean about initially 4,500 cells/cm2 from Banked hUC-MSCs?

b.      hUC-MSC sheet fabrication section: it is not clear the number of cells counted by the hemocytometer.

c.      Please add the reason for the Cell sheet tilting.

d.      Please add sequences of primers

3.      Result

a.      Please add figure S1; I could not find it

b.      Please report the unit of the centrifuge by g, not RPM

c.      What is your reason for choosing four days to analyze?

d.      (126 ± 7.0x104 cells) is not clear

Author Response

Response to review and revisions.

We thank the reviewers for their comments. These were very helpful and have greatly improved our manuscript. The following detailed responses address the reviewer’s comments:

Point 1: Please add references applying the power of centrifuge to improve adherence of cells.

Response 1: I added one sentence describing previous applications of centrifugal force to improve cell adherence with references (34-37) (page 2 lines 74-78).

Point 2: Why is necessary to increase the number of the cell sheet layer.

Response 2: We appreciate the reviewer’s question regarding the importance of controlling the number of cell sheet layers. In clinical applications of MSCs by systemic injection, 109 cells are generally administered with the expectation that <1% will be retained in circulation within 24 hours. The range for what constitutes a therapeutic cell dose is therefore unclear, particularly in the case that cells can be engrafted and localized at the target site. Having the capacity to increase the number of cell sheet layers allows for incremental control of the administered cell dose. Future in vivo studies will compare 1-, 2-, 3-, etc. layered sheets for therapeutic efficacy, allowing for a clearer determination of the therapeutic dose range that can be expected with engrafted MSC sheet therapy.

Additionally, an important conclusion from our study was that cell-cell interactions at the interface between layered cell sheets resulted in augmented pro-regenerative cytokine production per MSC. The technique for cell sheet layering described in this study provides a novel means of introducing discrete cellular interactions within 3D tissue that can accelerate MSC functionality. I elaborated this point in the conclusion section (page 11 lines 491-493).

Point 3: What did you mean about initially 4,500 cells/cm2 from Banked hUC-MSCs?

Response 3: I modified the sentence to clarify that this is the seeding density used for initiation from thaw and for expansion (page 3 lines 89-90).

Point 4: hUC-MSC sheet fabrication section: it is not clear the number of cells counted by the hemocytometer.

Response 4: I modified the sentence to express that at passage, the number of P6 cells harvested were counted by a trypan blue exclusion assay (page 3 line 99). Counting the cells at passage was necessary to achieve the proper seeding density onto the TRCD, 41,580 cells/cm2, described in the subsequent sentence.

Point 5: Please add the reason for the Cell sheet tilting.

Response 5: We incorporated methods for cell sheet tilting because in previous tests, we found that excess medium at the cell sheet-to-surface interface inhibited stable cell sheet adherence upon centrifugation. The data are included in the Supplemental section, pages 20-23. We also included one sentence to clarify the reason for cell sheet tilting in the Methods section (page 3 lines 108-109).

Point 6: Please add sequences of primers.

Response 6: We greatly appreciate the reviewer’s comment. The primers used in this study were all commercially obtained through Applied Biosystems, and therefore the sequence is not made publicly available without secondary sequencing of the amplicon. We included the TaqMan Assay ID for each primer in the Methods section, which corresponds with RefSeq data, the probe exon location, and the amplicon size as genomic reference. We included the explanation in the manuscript (page 5 lines 194-195).

Point 7: Please add figure S1; I could not find it.

Response 7: I included the Supplemental section, pages 20-23.

Point 8: Please report the unit of the centrifuge by g, not RPM.

Response 8: We have considered the reviewer’s valuable comment and converted all RPM values to relative centrifugal force (g-force) values based on the 16 cm centrifuge rotor.

Point 9: What is your reason for choosing four days to analyze?

Response 9: In unpublished studies, we determined that for cell sheets comprising around 1 million cells, viability is negatively impacted beyond 4 days of consecutive culture without medium exchange due to nutrient depletion and metabolite production. Our experiment focused on cumulative cytokine production by the cell sheet over time, and therefore medium exchange was not feasible because soluble cytokines would be removed. We therefore stopped the experiment at 4 days to avoid detriment to cell viability that could have confounded the results.

Point 10: (126 ± 7.0x104 cells) is not clear.

Response 10: 126 x104 ± 7.0x104 cells, page 8 line 329.

Reviewer 2 Report

This study demonstrates that layering cell sheets by centrifugation increases cytokine production, and is interesting for reader of biomedical field. While the experiments are well done, there are some points that need to address prior to my recommendation of acceptance.

1.       There is no supporting information (Figure S1) anywhere.

2.     Table 2: The viable cell number of 2-layers conventional cell sheet should be described.

3.     Figure 4 and 6: In Figure 4, the calculation of the relative expression of genes uses single cells as controls, but in Figure 6, conventional is used as a control. It is easier to compare the two if they are unified. Otherwise, only the relative expression levels of laminin, for example, would look very different and confusing between one-layer and two-layer centrifugation.

4.     Figure 5: Authors have mentioned that the decrease of MSC viability in conventional 2-layered cell sheets corresponds with up-regulation of HIF-1α expression, but HIF-1a is a gene expressed during hypoxia, but its expression is not always associated with cell survival. In particular, it has been reported that hMSCs retain the ability to thrive in even prolonged hypoxia, and hypoxia may be an essential component of the in vivo hMSC niche. (For example, WL Grayson et al J Cellular Physiol, 207, 331, 2006: Y Jin et al BBRC, 391, 1471 2010)

5.      In addition, authors have discussed the thickness of cell sheets and the diffusion of oxygen. Reviewer agree with this discussion, but doubt that oxygen does not reach a cell sheet with a thickness of 54 ± 3.1 μm but does reach a cell sheet with a thickness of 44 ± 4.8 μm. I doubt there is a clear cut-off value at the level of a few microns. Is there any data that shows that the cells are dead further away from the gas-liquid interface (at the bottom side) in these thickness of cell sheets? I'm not sure but my impression is that Oxygen seems to diffuse less easily in dense cell sheets. Is the effect of cell sheet density relevant?

6.    Figure 7e-g: Should be normalized by the numbers of cell.

Author Response

Response to review and revisions.

We thank the reviewers for their comments. These were very helpful and have greatly improved our manuscript. The following detailed responses address the reviewer’s comments:

Point 1: There is no supporting information (Figure S1) anywhere.

Response 1: I included the Supplemental section, pages 20-23.

Point 2: Table 2: The viable cell number of 2-layers conventional cell sheet should be described.

Response 2: The table was updated to reflect the reviewer’s comment, page 7. Due to cell death in the 2-layer conventional cell sheet sample that was measured at day 1, these cell sheets would detach in culture by day 3 and were therefore not able to be cultured until 4 days and measured for cell number. For this reason, we reported “not measurable” for 2-layer conventional cell sheets at the 4-day time point, explained on page 4-5 lines 184-186.

Point 3: Figure 4 and 6: In Figure 4, the calculation of the relative expression of genes uses single cells as controls, but in Figure 6, conventional is used as a control. It is easier to compare the two if they are unified. Otherwise, only the relative expression levels of laminin, for example, would look very different and confusing between one-layer and two-layer centrifugation.

Response 3: We appreciate the reviewer’s valuable comment and can agree that where possible, figures showing relative gene expression levels should be normalized to the same control group for direct comparison. In this case, the controls were selected for experimental purpose. Figure 4 shows that 1-layer cell sheets have higher cellular interactions (β-catenin, connexin 43, integrin β1) and cytokine production ability (VEGF, HGF, IL-10) compared to single cell suspensions, and this is enforced by 1-layer cell sheet centrifugation. Figure 6 shows that centrifugation cell sheet layering resulting in higher cellular interactions and cytokine production ability compared to conventional cell sheet layering. The distinctions are reported in the Figure Captions.

Point 4: Figure 5: Authors have mentioned that the decrease of MSC viability in conventional 2-layered cell sheets corresponds with up-regulation of HIF-1α expression, but HIF-1a is a gene expressed during hypoxia, but its expression is not always associated with cell survival. In particular, it has been reported that hMSCs retain the ability to thrive in even prolonged hypoxia, and hypoxia may be an essential component of the in vivo hMSC niche. (For example, WL Grayson et al J Cellular Physiol, 207, 331, 2006: Y Jin et al BBRC, 391, 1471 2010).

Response 4: We appreciate the reviewer’s highly considered response and agree that abundant evidence suggests MSCs are capable of hypoxic culture because they are derived from a hypoxic niche and that in some cases, hypoxia can enhance MSC functions. We removed the explanation that HIF-1α expression is related to cell death in the Result section (page 8, lines 312-314). We also modified the sentence about HIF-1α expression in the Discussion section (page 10, lines 431-435).

Point 5: In addition, authors have discussed the thickness of cell sheets and the diffusion of oxygen. Reviewer agree with this discussion, but doubt that oxygen does not reach a cell sheet with a thickness of 54 ± 3.1 μm but does reach a cell sheet with a thickness of 44 ± 4.8 μm. I doubt there is a clear cut-off value at the level of a few microns. Is there any data that shows that the cells are dead further away from the gas-liquid interface (at the bottom side) in these thickness of cell sheets? I'm not sure but my impression is that Oxygen seems to diffuse less easily in dense cell sheets. Is the effect of cell sheet density relevant?

Response 5: We appreciate the reviewer’s valuable comment and agree that 54 μm and 44 μm thicknesses may not be different enough to draw a clear diffusion limitation. The important data was the thickness of the layered cell sheets 1 hour after fabrication because the thickness in this early stage would affect cell death and HIF-1a expression measured after 24 hours of incubation. The thickness of 2-layer conventional cell sheets and 2-layer centrifugated cell sheet were 84 µm and 50 µm, respectively, after 1 hour of incubation. The thickness of the 2-layer centrifugated cell sheet was 0.6-fold reduced compared to the 2-layer conventional cell sheet. It was previously reported that five-layer human skeletal muscle myoblast (HSMM) sheets approximately 70-80 µm thick showed signs of hypoxia that three-layer HSMM sheets (approximately 40-50 µm thick, 0.5-0.6-fold) did not. These previously reported cell sheet hypoxic thresholds show comparability with our present findings. I changed Figure 5e from 24 hours data to 1 hour data to clarify the effects of the thickness in early stage (page 8, lines 308-311), as well as the Discussion section (page 10 lines 435-444).

Point 6: Figure 7e-g: Should be normalized by the numbers of cell.

Response 6: We have considered the reviewer’s valuable comment and agree that Figure 7 should normalize for cell number. Figure 7e-g now reports cytokine production per viable cell number. We also updated the Results section and Figure 7 caption to reflect this change.  

Round 2

Reviewer 1 Report

This manuscript is accepted in its present form.